# Sex estimation techniques based on skulls in forensic anthropology: A scoping review

**Xindi Wang[1], Guihong Liu[1], Qiushuo Wu[1], Yazi Zheng[1], Feng Song[1]\*, Yuan Li[2]\***

**1** Department of Forensic Genetics, West China School of Basic Medical Sciences & Forensic Medicine, Sichuan University, Chengdu, Sichuan, PR China, **2** Department of Forensic Pathology, West China School of Basic Medical Sciences & Forensic Medicine, Sichuan University, Chengdu, Sichuan, PR China

\* fengsong9@163.com (FS); 291846740@qq.com (YL)

## Abstract

### Background

Sex estimation is an essential topic in the field of individual identification in forensic anthropology. Recent studies have investigated a growing range of techniques for estimating sex from human skulls.

### Objectives

This study aims to provide a scoping review of the literature on techniques used in skull-based sex estimation, serving as a valuable reference for researchers.

### Sources of evidence

The literature search was performed using PubMed, Scopus, and Web of Science from January 2020 to February 2024.

### Eligibility criteria

Eligible studies have investigated issues of interest to forensic anthropology about sex estimation using skull samples.

### Charting methods

A total of 73 studies met the inclusion criteria and were categorized and analyzed based on the anatomic sites, modalities, trait types, and models. Their accuracy in estimating sex was subsequently examined, and the results were charted.

### Results and conclusions

Our review highlights that the 3D medical imaging technique has enhanced the efficiency and stability of skull-based sex estimation. It is anticipated that advancements in 3D imaging and computer vision techniques will facilitate further breakthroughs in this field of research.

**Data Availability Statement:** All relevant data are within the manuscript and its Supporting Information files.

**Funding:** This study was supported by the National Natural Science Foundation of China (No.

82202079) and Natural Science Foundation of Sichuan Province (No. 2022NSFSC1403), both awarded to Ms Yuan Li. The funders had no role in study design, data collection and analysis, decision to publish, or preparation of the manuscript.

**Competing interests:** The authors have declared that no competing interests exist.

## Introduction

Forensic anthropology often involves estimating sex, ancestry, age, and height from the skeletal remains of unknown individuals. This estimated summary is referred to as the biological profile, which is subsequently used to compare with missing person records to achieve personal identification [1]. Successfully estimating sex is a prerequisite for developing a reliable biological profile because the estimation of stature, age, and ancestry follows patterns related to sex [2].

With regard to human skeletons, morphological variations have been attributed to diverse factors. These include but are not limited to, the unique developmental pathways specific to each sex, hormonal responses to environment stimulation, and the adaptability shaped by sexual labour division in past societies [3–5]. These factors contribute to the inherent variations in the appearance between male and female skeletons, thereby enabling skeletal structures to be informative indicators in sex estimation [6]. Numerous published studies investigate various skeletal features and their capacity to differentiate between sexes effectively. These features encompass, but are not limited to, the pelvis [7–9], skull [2, 4, 10], ribs [11, 12], sternum [13–15], vertebrae [16–18], clavicle [19–21], and limb bones [22–24]. Among these, the pelvis is referred to as the most reliable bone for sex estimation [7, 25, 26]. Nevertheless, the skull is considered the most reliable alternative structure when the pelvis is unavailable for analysis or displays ambiguous signals [27].

The skull, comprising both the cranium and the mandible, presents an interesting set of traits for studying morphological variation in relation to the sexes or other factors, such as genetics, developmental processes, dietary habits, and environmental influences [4, 13]. On average, the male skull is larger and thicker, whereas the female skull is smaller and smoother [4, 28, 29]. Sexual dimorphism in cranial morphology is evident not only in relative size but also in distinct shape features. For instance, males tend to develop a more prognathic (with a protruding jaw) and dolichocephalic (elongated head shape) cranium compared to females [30, 31]. As per recent research, the shape of the skull, particularly the cranial and facial bones, is more likely to provide potentially useful information for sex estimation [4].

Sex estimation based on skulls involves three fundamental factors: modalities, methods, and analysis models. Fig 1 illustrates the process of sex estimation for better understanding.

The orange box illustrates the skull modalities available for sex estimation, including dry skulls, radiographs, Computed Tomography (CT), and Magnetic Resonance Imaging (MRI) scans, as well as three-dimensional (3D) models derived from CT reconstruction or 3D scanning. The blue box outlines non-metric, traditional metric, and geometric morphometric (GM) methods for sex estimation. Within GM methods, there are further subdivisions into landmark-, outline-, and surface-based approaches. The green box shows analytical models utilized for sex estimation. Statistical analysis includes discriminant function analysis and logistic regression algorithms, while classical machine learning involves multiple algorithms, with support vector machine, neural network, decision tree, and random forest presented in the diagram as representative examples. Deep learning, featured in both blue and green boxes, stands out for its ability to automatically extract features and classify sexes. By combining different modalities, methods, and models, a broad range of sex estimation outcomes can be achieved.

## Modalities

Forensic anthropologists have traditionally preferred to obtain primary data from skull collections. The drawback of metric methods based on skull collections is that the manipulation may cause secondary damage to the bone. Fortunately, the three-dimensional (3D) scanning technique addresses this issue by providing digital 3D models of dry skulls.

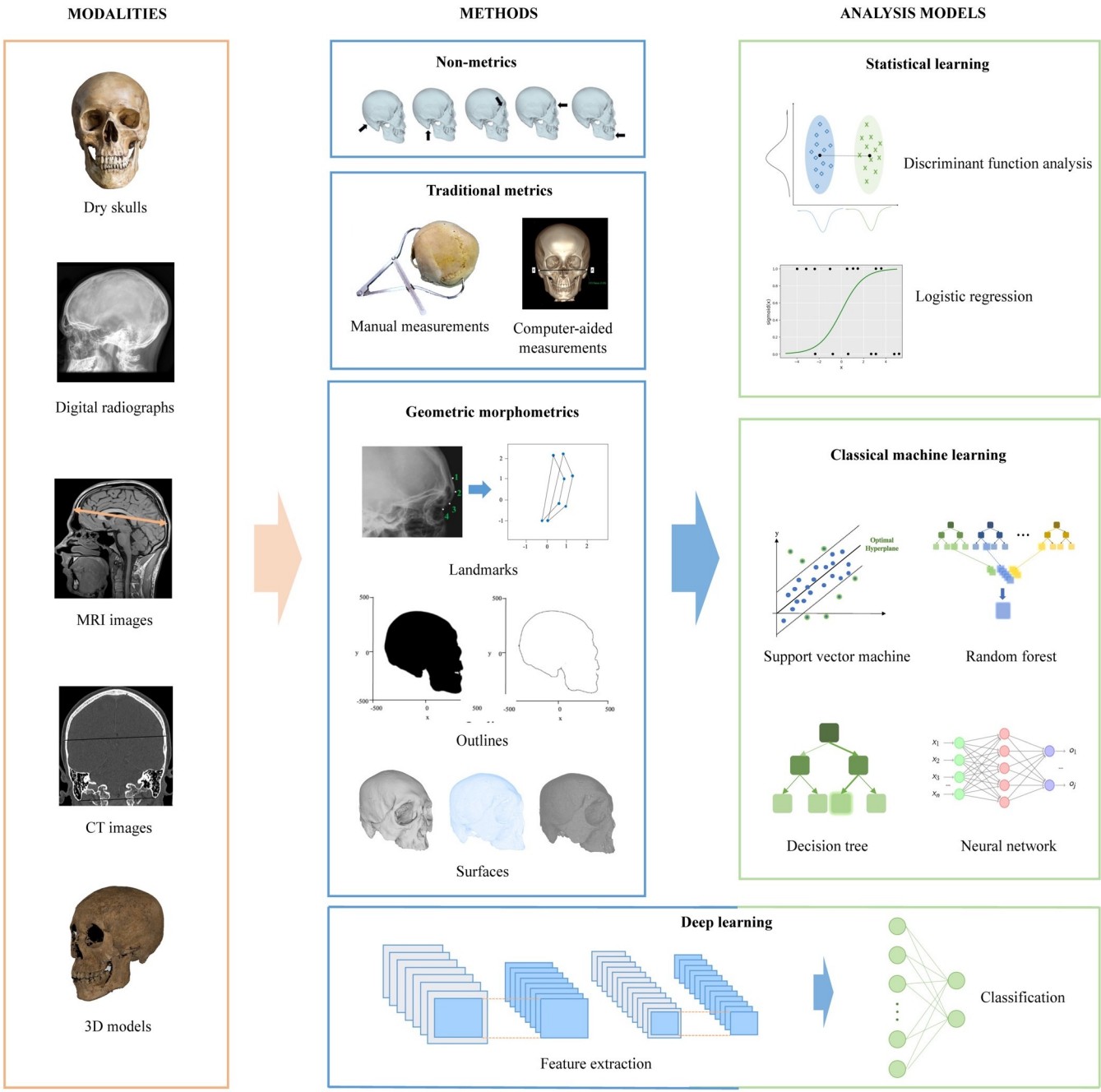

**Fig 1. Overall framework of sex estimation based on skulls.**

Digital radiography provides architectural and morphological details of the skull and thus reveals multiple anatomical landmarks for comparison, making it ideal for sex estimation in charred or decomposed cases [32, 33]. However, radiographs present the 3D characteristics of the skull on a two-dimensional (2D) image. Thus, the morphometric traits of the skull superstructures and intracranial structures sometimes appear ambiguous on a 2D roentgenogram [34].

The Computed Tomography (CT) images from multi-planar reformation and Magnetic Resonance Imaging (MRI) scans offer high-resolution views of the targeted plane and the

internal edges of the skull, thereby facilitating precise positioning of landmarks and measurements [35]. However, these sectional images still represent the cranial morphology at a 2D level, posing difficulties in accurately measuring the cranial surface. The CT reconstruction using the volume rendering technique offers a significant opportunity to overcome the limitations of 2D imaging by modeling skull collections at the 3D level. These 3D models provide a more comprehensive and detailed understanding of cranial morphology [35–38]. With the aid of computer visual techniques, 3D models hold the potential to reveal more cranial traits, further enhancing the accuracy of sex estimation.

## Non-metric and metric methods

Sex estimation methods are generally classified into non-metric (morphologic) and metric [3]. Morphological methods rely on visual assessment of the various structures and configurations of skulls. Walker's method [39], which relies on standard morphological traits originally described by Buikstra and Ubelaker [40], is one of the most widely used morphological methods for estimating sex from the skull. This standard comprises five traits: nuchal crest prominence, mastoid process size, supraorbital edge thickness, supraorbital ridge/glabella prominence, and mental eminence. Each variable is scored ordinally on a scale ranging from 1 (most gracile) to 5 (most robust). Walker [39] developed this standard using samples from North America and Britain, achieving an accuracy rate of 89%. These non-metric traits allow for rapid preliminary sex estimation results without using any special equipment; however, they are inherently observer-dependent and necessitate substantial scoring experience for cranial features [36, 41, 42].

Metric evaluation is founded on the quantification of sexual dimorphism in skull dimensions, yielding more objective results for sex estimation [43, 44]. Nevertheless, it often necessitates precise localization of anatomical landmarks. Metric methods could be further divided into traditional metrics and geometric morphometrics (GM). Traditional metric methods develop statistical models based on cranial indicators derived from manual or computer-aided measurement techniques, such as linear distances, angles, areas, and indices.

GM enables the morphology quantification of rigid structures of skulls and the comparison of the size and shape separately between males and females. The anatomical landmarks, outlines, and homologous surfaces are employed to capture the geometric differences in sexually dimorphic structures. Landmark-based geometric morphometric approaches typically employ landmarks as Cartesian coordinates to quantify the shape difference of skulls. Generalized Procrustes analysis (GPA) is most commonly used to conduct the landmark configuration superimposition, in order to eliminate the differences in the location, scale, and size across configurations. After landmark standardization, the consensus mean configurations for males and females will be generated. Following that, shape variation among superimposed configurations between the target skull and the mean male/female shape can be quantified as Procrustes distances [3, 45]. These shape variation parameters provide objective indicators to estimate the sex of the target skull.

Compared with landmarks, outline-based methods apply landmarks to represent contours or boundary outlines. They are independent of the presence of true anatomical landmarks, thus providing a more complex, quasicontinuous representation of sex variation. As a curve-fitting function, Fourier analysis decomposes spatial information about an outline into an infinite series of sine and cosine functions, weighted by Fourier descriptors that represent shape variables [46].

The surface-based approach involves employing homologous models with numerous vertices to represent or approximate the surface of 3D digital skulls. Surface registration techniques such as the iterative closest point and coherent point drift [47] are employed to mitigate

deviations caused by unequal location and rotation. The enormous amount of vertex data on the 3D skull surface is reduced to lower dimensions by principal component analysis before statistical processing. Although it doesn't rely on landmarking, analyzing massive point data demands robust predictive models and increased computational power.

## Analytical models

The analytical models for sex estimation can be roughly divided into statistical learning, classical machine learning (ML), and deep learning (DL), with the latter two collectively referred to as artificial intelligence (AI).

Statistical learning includes discriminant function analysis (DFA) and logistic regression (LR), which is the simple choice for sex estimation. DFA projects m-dimensional data into a space where the results have the properties of maximum discrimination between classes and minimal intraclass distance [48]. A linear function is formulated by combining a set of variables to calculate the classification score for each sample. This score is subsequently compared with the threshold determined from the group centroids [49]. LR predicts the probability of an event occurring by fitting a sigmoid function to the data and has been regularly employed in metric indicators for sex estimation [48].

Classical ML, such as the support vector machine (SVM), decision trees (DT), and neural networks (NN), have been used for skull-based sex estimation. Unlike statistical techniques that operate solely on group parameters, such as means and covariance matrices, classical ML algorithms are capable of creating nonlinear decision boundaries between groups with minimal knowledge of the domain or presumptions about the probability distributions underlying the observed data [50].

DL, composed of multiple neural network layers, allows for extraction and learning representations of training data with multiple levels of abstraction [51]. As one type of DL algorithm, the convolutional neural network (CNN) is designed to learn spatial hierarchies of features from visual data, such as images. The convolution layer is a fundamental component of CNN architecture that applies convolution kernels to extract features from each position on images [52]. Compared with statistical and classical ML learning, CNN can perform feature extraction automatically without hand-crafted involvement, due to the utilization of convolution layers. It has proven itself a powerful method in medical fields that rely on imaging data, including radiology, pathology, dermatology, and ophthalmology [22, 53–56]. However, CNN models are currently seldom employed in the field of sex estimation using skulls, which may be partly explained by the necessity for extensive training datasets and a profound level of technical expertise in AI.

## Objectives

The selection of modalities, methods, and analytical models for skull-based sex estimation is diverse. Consequently, studies in this field are vast and intricate, yielding a broad spectrum of outcomes. Our review aims to summarize the various modalities, non-metric and metric methods, predictive models, and software employed by recent studies, highlighting innovative techniques to enhance the reliability and accuracy of sex estimation.

## Materials and methods

### Protocol and registration

This systematic review followed the Preferred Reporting Items for Systematic Reviews and Meta-analysis Protocols extension for scoping reviews (PRISMA-ScR) [57]. The checklist of the PRISMA-ScR guidelines is provided in the S1 Table.

### Search process

The search was conducted using PubMed, Scopus, and Web of Science databases. The search string included keywords related to the study's aim: ('sex' OR 'gender') AND ('determination' OR 'estimation' OR 'prediction') AND 'skull'. Filters were applied to limit the search results to articles published between January 2020 and February 2024 in English and Chinese.

### Eligibility criteria

The following studies were included: (a) original articles; (b) studies published in Chinese or English; (c) studies using craniums, mandibles, or whole skulls for sex estimation. The exclusion criteria included the following: (a) non-human subjects; (b) conference papers, reviews, case reports, and books; (c) studies using other bones (including teeth) other than skulls; (d) intact or fractured skulls; (e) studies using samples of sub-adults under the age of 18 years old; (f) studies without disclosing the outcomes of their sex prediction models.

### Study selection

After the search was completed, duplicates were removed, and relevant articles were selected from the titles and abstracts. The inclusion and exclusion criteria listed for this systematic review were considered while reading the full text to ascertain eligibility. Two independent researchers (XW and GL) carried out the systematic search, study selection, and data extraction. Disagreements among researchers were resolved by consensus, mediated by a third reviewer (YZ).

### Data extraction

Information about authors, reference numbers, publication years, population, anatomical site, sample size, modalities, analysis models, inter/intra bias, and predictive accuracy were extracted for each included study.

## Results

### Selection of sources of evidence

Our preliminary search yielded 2200 records in total. Following the duplicates' elimination and automatic filtering, 716 studies remained. After reviewing the titles and abstracts, we excluded 507 articles from further consideration. A total of 136 full-text articles were excluded since they failed to meet the inclusion or exclusion criteria, resulting in a final inclusion of 73 studies in this scoping review. The procedure of the literature search is summarized in Fig 2. The specific characteristics of each study are summarized in the S2 Table.

### Synthesis of results

**Populations.** The majority of studies on sex estimation were conducted on Asian populations, with India [58–71] leading the way, followed by Turkey [35, 72–78], Iran [79–84], and China [85–87]. In Europe, notable contributions came from Bosnia [88–92] and Bulgaria [36, 93–96], while a single study encompassed four distinct Western European populations: Belgium, Switzerland, France, and Portugal [97]. Several studies were conducted in African countries, including Nigeria [98–100] and Egypt [101, 102], and in American countries, including Brazil [103, 104] and the United States [105]. Notably, one study concerned Caucasians, but the countries where this study was conducted were not specified [37]. Fig 3 shows the detailed distribution of the studied population.

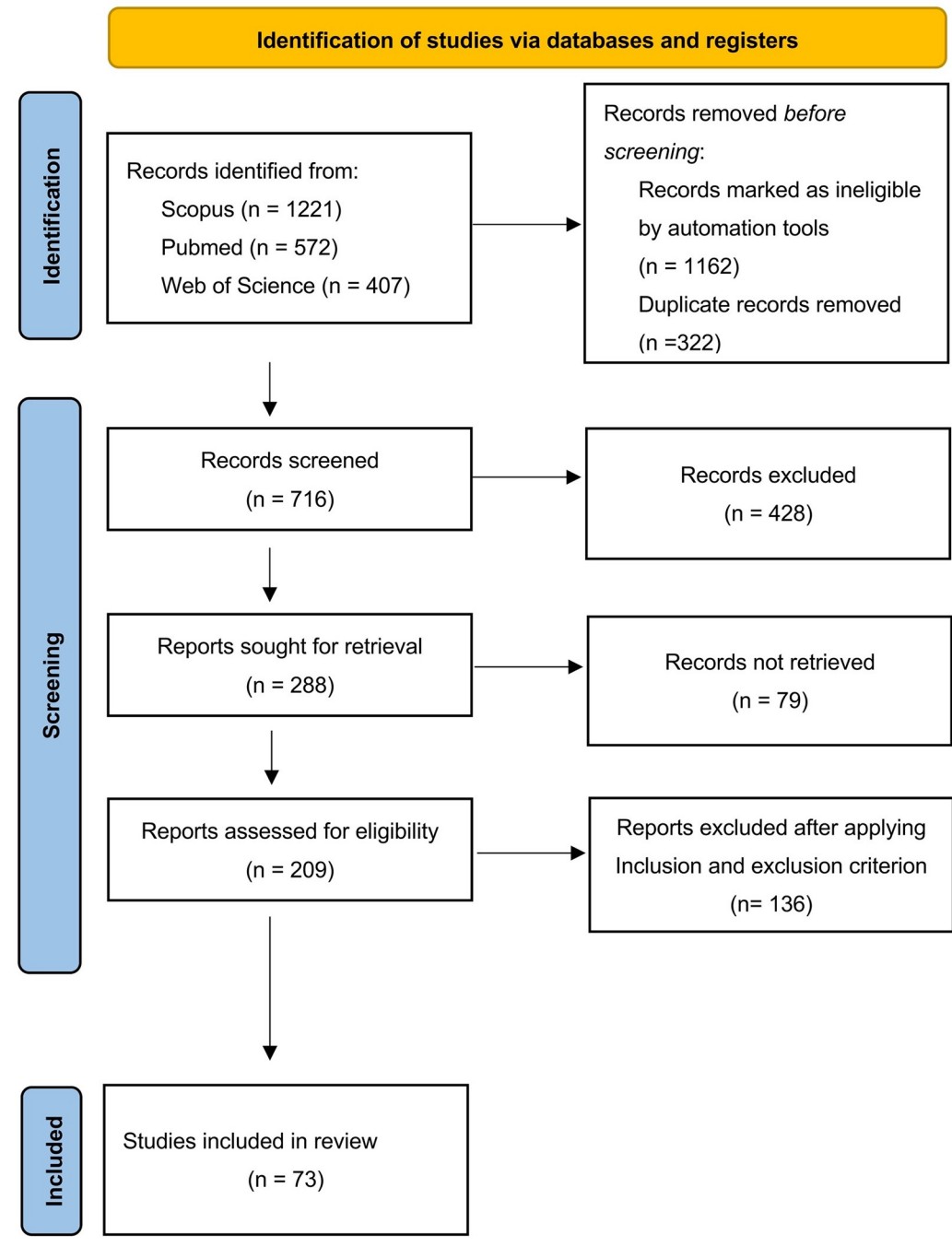

**Fig 2. PRISMA flow of study selection process.**

Fig 3 depicts the geographical distribution of the populations investigated across 73 studies, where darker shades of blue signify a greater frequency of research conducted within the respective region.

**Modalities.**   Among the included studies, CT was the most frequently employed imaging modality, particularly in the form of 3D skull models derived from CT reconstructions (Fig 4A). Moreover, one study conducted by Kumar et al. utilized both 2D CT scans and 3D

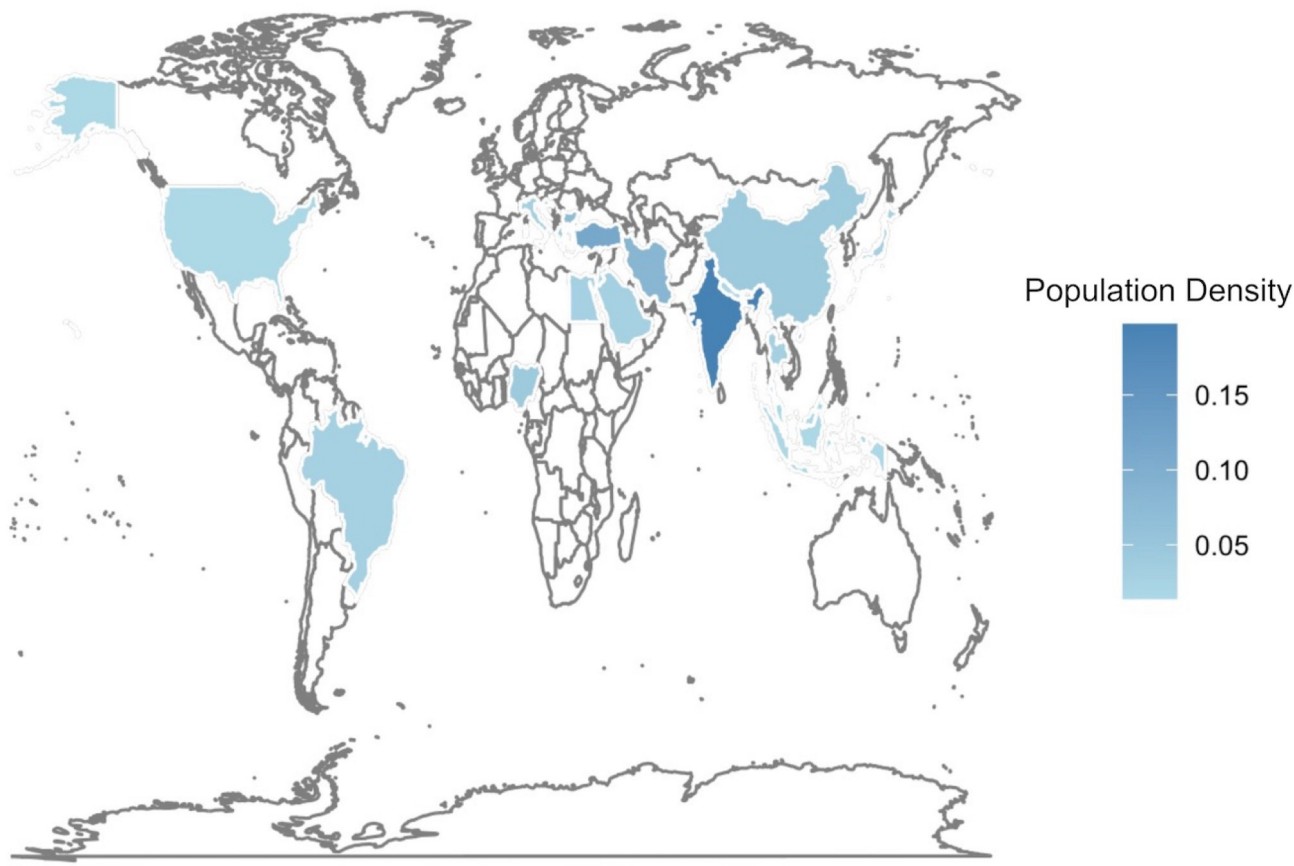

**Fig 3. World map with population density.**

CT models of 217 Indian skulls for metric analysis, achieving an accuracy rate of 93.6% [63]. In contrast, MRI technology was solely employed in the study by Liu et al. [85], where they assessed sexual dimorphism in the sphenoid sinus of Chinese individuals, albeit with a relatively low accuracy of 63.3%.

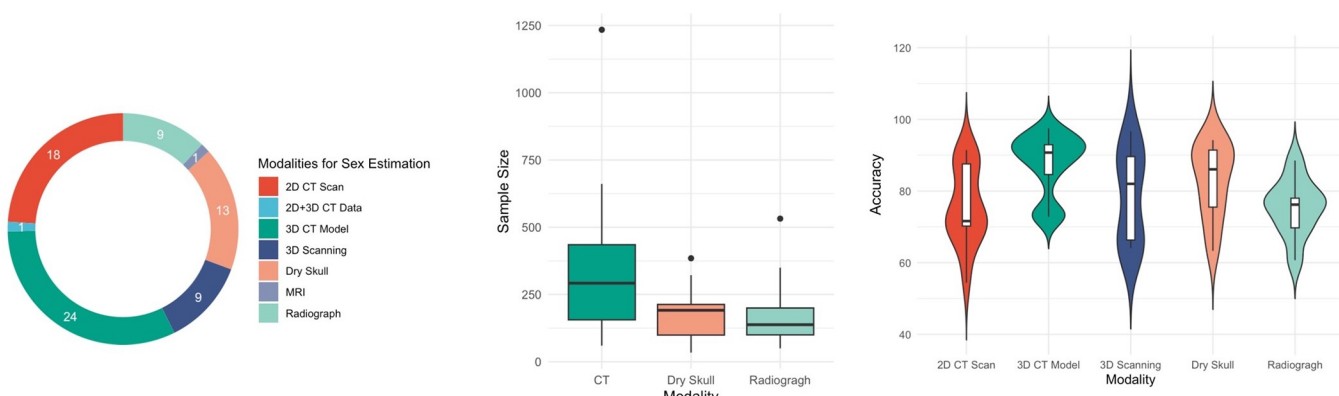

**Fig 4.** (a) Numbers of included studies using different modalities for sex estimation; (b) Sample sizes of studies using different modalities; (c) Accuracy corresponding to each type of modality.

The sample sizes in these studies varied significantly, ranging from 34 to 1234 cases. Notably, the sample sizes for CT data tended to be larger than those for dry skulls and radiographs (Fig 4B). The study with the largest sample size came from Kondou et al. [106], who utilized CNN algorithm to conduct sex estimation on 3D CT skull data from 1234 Japanese individuals. This resulted in exceptionally high accuracy rates of 95.0% and 93.0% for the training and testing datasets, respectively.

The accuracy of 3D models derived from CT reconstruction was consistently high and outperformed that of other modalities (Fig 4C). Despite the fact that five studies focusing on sex estimation in partial skull regions reported accuracies falling below 75% [74, 107–110], the majority of studies that employed 3D CT models consistently achieved accuracies exceeding 85%. In terms of dimensionality, both CT slices and radiographs, being 2D planar images, exhibit reduced accuracy in sex estimation compared to 3D models.

**Non-metric and metric methods.** The majority of studies utilized traditional metric traits, followed by GM and non-metric methods. Additionally, one study by Jeong et al. [107] employed both traditional metric and non-metric methods for sex estimation (Fig 5A). Notably, the study by Koudou et al. [106] based on CNNs was excluded, as the network automatically extracted the cranial features, falling outside the categories of metric or non-metric traits.

To decrease the number of variables, we focused our comparison on studies that employed 3D skulls as the modalities, including dry skulls, digital models from 3D scanning, and 3D CT models. Among the studies we reviewed, all non-metric [76, 87, 104, 111–115] and GM studies [36, 84, 86, 88–92, 94–96, 109, 110, 116, 117] and 19 out of the 47 metric studies employed 3D cranial models. Additionally, the only study incorporating metric and non-metric approaches also utilized 3D models for its analysis [107]. All three methods exhibited a broad range of accuracy levels, yet their accuracy rates' differences appeared ambiguous (Fig 5B).

Among traditional metric studies, we identified nine partial skull regions that were investigated for sexual dimorphism, underscoring their significance in sex estimation (Fig 5C). Notably, the foramen magnum (FM) and occipital condyles (OC) emerged as the most frequently examined anatomical landmarks, closely followed by the mandibles and mastoid processes. Nevertheless, despite their prevalence, neither FM nor OC achieved a high level of accuracy, failing to exceed 80% in distinguishing between males and females. In contrast, Farhadian et al. [79] employed measurements derived from the mastoid process, reporting the highest accuracy rate of 97.5% for sex estimation.

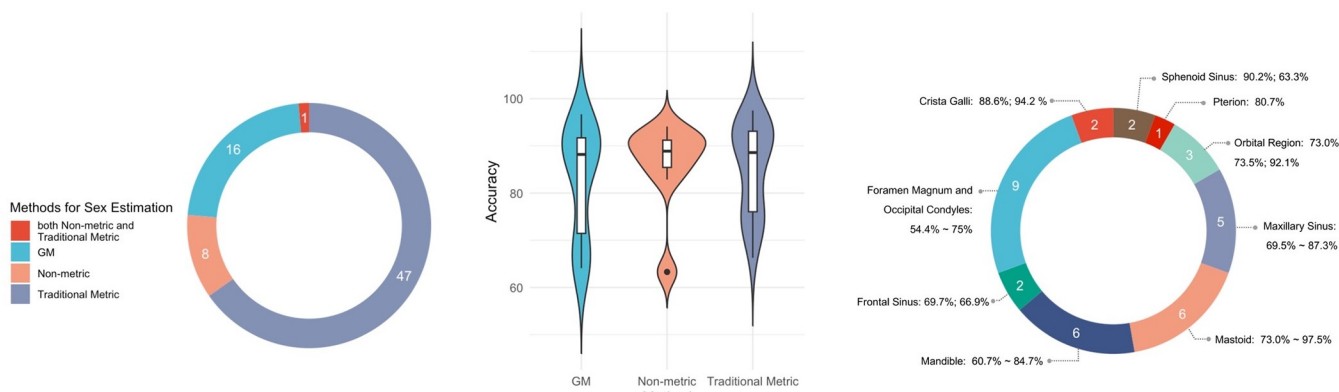

**Fig 5.** (a) Numbers of studies using different methods for sex estimation; (b) Accuracy distribution corresponding to each method on 3D cranial models; (c) Anatomical regions used by traditional metric studies. The rings are labelled with the names of anatomical sites and the accuracy achieved in studies.

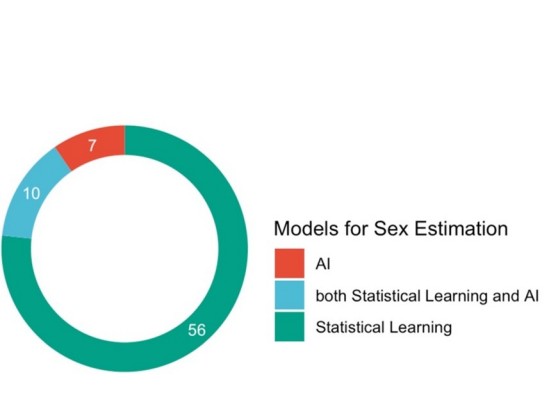
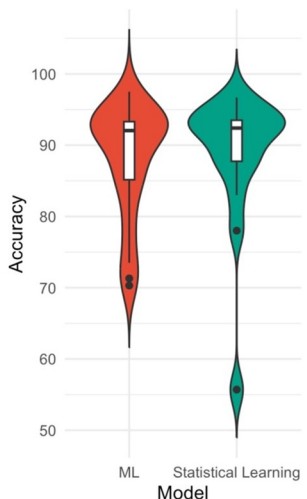

**Fig 6. (a) Numbers of included studies using different predictive models for sex estimation.** AI: Artificial Intelligence; **(b) Accuracy corresponding to each type of analytical model.** ML: Machine Learning.

**Analytical models.** Seventy-six studies employed statistical learning approaches, while seven solely utilized AI algorithms [36, 86, 101, 106, 110, 116, 118]. Moreover, ten articles incorporated both ML and statistical learning methods (Fig 6A) [75, 79, 83, 93, 96, 103, 111, 115, 117, 119]. This section specifically examined the accuracy of the models employed in these ten studies. Interestingly, when compared to simple statistical learning models, the accuracy of sophisticated machine learning models did not increase (Fig 6B). Furthermore, it is not feasible to assess DL's performance in this situation, given that only one study employed the DL algorithm [106].

## Software for skull-based sex estimation

From the included studies, we identified three software tools for sex estimation using skulls. The CroWalker software was developed by Bareša et al. [114] utilizing 200 MSCT scans from the modern Croatian population. This software employs four non-metric traits- nuchal crest, mastoid process, supraorbital edge, and glabella- to estimate sex based on LR equations. The scoring of these traits follows the guidelines in the MorphoPASSE user manual. CroWalker allows users to determine the probability of an individual being male or female, leading to a final sex classification. In Bareša et al.'s study [114], the best-performing equation, which included the glabella and mastoid process, achieved an accuracy rate of 86.25%.

The SexEst software, developed by Constantinou and Nikita [120], is accessible online at http://sexest.cyi.ac.cy/. This software utilizes 43 variables to measure cranial and postcranial bones, drawing upon the Goldman dataset (Auerbach & Ruff, 2004) and the Howells dataset (Howells, 1973; Howells, 1989; Howells, 1995). Additionally, SexEst integrates various classification models, including DFA, SVM, random forest (RF), and gradient boosting (GB), allowing users to choose the model that best suits their data. Nikita, P.A. et al. [119] conducted a study to assess the efficacy of SexEst on Greek individuals, and the light GB model achieved the highest accuracy rate of 73.5% for sex classification.

KKU Sex Estimation, developed by Techataweewan et al. [43], is available at https://tinyurl.com/26t3x3mu. This software is grounded on the dataset consisting of 322 Thai skull collections sourced from Khon Kaen University. Using the DFA model, it can handle 25 cranial and

5 mandibular measurements or only a single measurement. The output includes the predicted sex, measures of model success, and a histogram showing the position of the "Unknown" within the distribution of the reference samples. In the study by Techataweewan et al. [43], the accuracy rate of sex estimation was 92.1%.

## Discussion

A substantial number of studies have addressed the issue of sex estimation based on the features of human skulls between January 2020 and February 2024, demonstrating the significance of this topic in forensic anthropology. The development in medical imaging technology has significantly broadened the scope of modalities available for sex estimation research, such as X-rays, CT, and MRI scans. Both non-metric and metric approaches coexist and complement each other, revealing the sexual dimorphism in cranial morphology. Furthermore, AI algorithms offer more advanced and sophisticated model architectures for sex estimation, capable of analyzing the intricate data acquired through medical imaging technologies. This study overviewed the recent advancements and outcomes of skull-based sex estimation technology. Our findings underscore the effectiveness of 3D cranial models in enhancing the accuracy of sex estimation and emphasize the need to further explore the untapped potential of AI algorithms in sex classification.

### Population

Differences in skull structure among various populations greatly impact the accuracy of both metric and non-metric assessments for sex estimation. For instance, the study conducted by Techataweewan et al. [43] highlighted sexual dimorphic differences in skulls among Thai, American, and African populations. They measured 20 cranial metrics on 319 dried Thai skulls and compared the results to those reported by L'Abbé et al. for Americans and Africans. Across all craniometric characteristics, the average percentage differences between males and females were found to be 5.62% in Americans and 3.65% in Africans. The Thai population exhibited an intermediate difference of 4.78%. Furthermore, several non-metric studies indicated that Walker's sex estimation standard inadequately represents the sexual dimorphism presented by several populations, resulting in incorrect sex estimation for many individuals [76, 87, 112–114]. Consequently, expanding the investigated populations and implementing population-specific adjustments is imperative to account for the varying patterns of sexual dimorphism.

### Modalities

With a spurt of progress in medical imaging technology, the scope of materials available for sex estimation research has expanded. Medical imaging enables the broadening of databases from skull collections to larger hospital records, thereby encompassing characteristics from contemporary, dynamic populations. However, it is noteworthy that the use of hospital records simultaneously introduces ethical concerns surrounding the acquisition of medical records and data.

According to the literature we examined, CT is the most widely used medical imaging modality for skull-based sex estimation. Compared to X-rays, both CT and MRI provide great resolution without overlapping structural interference, but only one study used MRI technology [85]. In contrast to MRI, CT imaging has higher resolution and clarity for skeletal structures, allowing for clear observation of skull morphology, density, and subtle structural changes. This is particularly crucial for accurately measuring and analyzing skull features during sex estimation. Additionally, CT has higher scanning rates and is more cost-effective,

making it better suited for forensic investigation and identification. More importantly, CT reconstruction technology has greatly facilitated precise 3D skull modeling.

The advent of CT reconstruction using the volume rendering technique provides a significant opportunity to model the skull collections at the 3D level. The measurements of 3D models and skull collections are relatively comparable due to the high resolution of CT scans producing negligible differences [63, 93, 121]. Moreover, digital 3D models provide flexible observation views around the entire skull and facilitate point positioning for operators. Combined with the aid of computer measuring software, 3D models enable researchers to explore more traits on the cranial surface. Consequently, sex estimation can benefit from 3D models, which allow detailed observation, precise landmarking, and computer-aided measurement, thereby enhancing the applicability and accuracy of both qualitative and quantitative methods [36–38, 63, 93, 122].

## Non-metric and metric methods

The accuracy between non-metric and metric methods didn't seem to exhibit a notable difference. However, one particular study by Jeong et al. [107] concluded that the metric methods for sex estimation are more useful than non-metric methods. They applied both traditional metric and non-metric methods for sex estimation on the mastoid process, revealing that the metrics achieved an accuracy rate 20% higher than the non-metrics. Consequently, the further comparison between metrics and non-metrics necessitates the standardization of variables, such as the same sample size, anatomical site, and predictive algorithm.

While traditional metric traits primarily reflect cranial size, GM methods offer a distinct perspective by capturing shape variations between male and female skulls. A study by Ajanovic et al. [89] employed GPA to investigate sex differences in the shape of orbital regions among Bosnians, exhibiting an average accuracy rate of 87.6%. This result is comparable to the 92.1% accuracy reported by Packirisamy et al. [123], who employed traditional metric indicators based on Saudi Arabian skulls.

In another study, Imaizumi et al. [116] created homologous surface models of the entire skull, cranium, and mandible using 100 skull shapes derived from CT reconstructions. They applied partial least squares regression to reduce dimensionality and generated principal components (PCs) for each model. When the first PC was input into an SVM classifier, the following accuracy rates were achieved: 90.6% for the whole skull, 90.7% for the cranium, and 84.1% for the mandible. The surface-based outcome mirrored the findings of several previous studies that employed traditional metric methods, indicating the efficacy of GM methods [36, 69, 94–96].

In terms of traditional metric traits, the bizygomatic breadth (BZB) has been found in studies from Turkey [73], Thailand [43], Iran [83], India [63, 68], Greece [119], and Caucasian populations [37] to be one of the most reliable predictors of sex, with an accuracy rate ranging from 77.2% to 89.2%. Additionally, as presented in relevant studies, the metric traits from partial regions of the skull could potentially be utilized for sex estimation.

**Mandible.** Denny et al. [59] employed mandibular width for sex estimation among Indians, achieving an accuracy rate of 68.4%. However, Sain et al. [69] achieved a higher accuracy of 80% by utilizing the gonion-gnathion length derived from Indian mandibles. The ramus and chin height exhibited significant sexual dimorphism in both Rad et al.'s study [81] for Iranians and Cappella et al.'s study [124] for Italians. In addition to linear indicators, angular measurements of the mandible have also been studied for sex estimation. The condylar angle and notch angle were combined by Abualhija et al. [125], successfully classifying 77.6% of Jordanian samples. Meanwhile, Girdhar et al. [60] solely focused on the gonial angle and achieved a lower accuracy rate of 60.7% for sex estimation in the Indian population.

**Mastoid.** The distance between the most prominent convex mastoid point and intermastoid distance were validated in the Iranian population, achieving an impressive accuracy rate of over 90% in two studies [79, 82]. In contrast, when measuring mastoid volumes in Korean individuals using 3D CT models, Jeong et al. [107] found a lower accuracy rate of 73.0%, similar to Petaros et al.'s 74.1% accuracy in Croatians [109]. Additionally, two Indian studies revealed 76.7% and 78.0% accuracy for mastoid height and area, respectively [66, 67].

**Foramen magnum and occipital condyles.** Meral et al. [35] and Aljarrah et al. [126] reported that FM area was the most accurate single predictor, achieving an accuracy of 74.5% and 64.1%, respectively. In three separate studies [99, 102, 127], the FM width emerged as a significant sex difference, consistently achieving an accuracy of approximately 65.4%. In a Malaysian study, Soon et al. [128] obtained 60% accuracy using FM's transverse and anteroposterior diameters. Additionally, the maximum length of the right OC was found to be significant for sex estimation in three investigations [99, 102, 127], with accuracy ranging from 58.9% to 66.5%. Notably, when combining the FM and OC predictors, a slight enhancement in the performance of sex estimation was observed in three studies [102, 126, 127].

**Paranasal sinus.** Four studies employed 2-D CT scans and consistently reported that the maxillary sinus (MS) height was the most effective variable in indicating sexual dimorphism [58, 62, 65, 100]. Additionally, Kurniawan et al. [64] calculated the MS index from Indian lateral cephalograms, attaining an accuracy of 76.2%.

Banihashem et al. [80] analyzed the sphenoid sinus volume (SSV) using 2-D CBCT images from 469 Iranian individuals, demonstrating that the SSV could accurately distinguish sex with an average accuracy of 90.2%. In contrast, Liu et al. [85] achieved a lower accuracy of 63.3% when utilizing the SS area calculated from the midsagittal view of 73 Chinese MRI scans.

Two studies evaluated the effectiveness of the frontal sinus (FS) in sex estimation, achieving an approximate accuracy of 68.3% [50, 77]. They reported FS height, width, and anteroposterior diameters as useful indicators of sexual dimorphism.

**Other anatomical regions.** Two studies indicated that the orbital traits had limited forensic value for sex classification in Croatian and Turkish populations, achieving an accuracy rate of approximately 73.0% [74, 108]. Conversely, Packirisamy et al. [123] demonstrated that the orbital region exhibited high potential for sex estimation among Saudi Arabians, achieving a high 92.1% accuracy rate. The orbit height was the most reliable variable, with an accuracy rate of 83.8%.

In terms of the crista galli (CG), Komut et al. [72] found that the CG length on axial slices of CT was the most effective variable, achieving an accuracy of 83.7%. Golpinar et al. [78], on the other hand, concluded that CG height was the most predictive factor, achieving an accuracy of 88.4%. Both studies underscored the significance of CG measurements in obtaining reliable sex classification results.

Furthermore, a study conducted by Uabundit et al. [118] on 124 dry skulls from Thailand suggested that pterion measurements might hold potential for sex estimation, achieving an accuracy of 80.7% in classification.

## Analytical models

In the context of sex estimation, complex classical ML models do not necessarily achieve better accuracy. Sex estimation is a straightforward binary classification task. For this reason, simple statistical learning models are often capable of yielding satisfactory results. Consequently, when complex AI models were employed for sex estimation, they did not significantly enhance the accuracy rates. Additionally, the comparison between statistical models and AI algorithms

was hampered by a limited number of studies (only 10 in total). This constraint might be another reason why the difference in accuracy between the two types of analytical models was relatively small.

Among the reviewed articles, four studies reported their accuracy rates for sex estimation exceeding 95%. Toneva et al. [93] conducted a comparative analysis of three models—SVM, NN, and LR—and found that SVM emerged as the most effective classifier, achieving an accuracy of 96.1%. In a separate study, Farhadian et al. [79] developed 11 models based on measurements from Iranian mastoids. Notably, the RF model achieved the highest accuracy rates, with 97.5% for training data and 96.9% for testing data, while statistical models lagged behind with an accuracy of 92.5%. Utilizing GM methods, Bertsatos et al. [117] generated elliptic Fourier descriptors from the 2D projection of the nasion–bregma ectocranial segment, which achieved an accuracy of 96.7% in the DFA model. A specific study by Kondou et al. [106] implemented a CNN algorithm using voxel data from 3D cranial models, achieving impressive accuracies of 95.0% and 93.0% for training and testing datasets, respectively.

These four studies, each showcasing a unique algorithm achieving high accuracy in sex estimation, underscore the variability in performance among different models. Therefore, the multi-angle evaluation of various algorithms is crucial for selecting the most efficient classifier [50]. By evaluating algorithms from multiple angles, researchers might gain a comprehensive understanding of their strengths and limitations, ultimately leading to more accurate and efficient predictive models.

Additionally, AI algorithms are often regarded as "black boxes," where their internal workings remain opaque to external scrutiny, raising ethical concerns regarding applying such algorithmic evidence in criminal justice proceedings. To address these concerns, future research is suggested to persist in exploring AI statistics for sex estimation, thus enhancing the interpretability of these algorithms [129, 130].

## Software for skull-based sex estimation

The development and open access of software have improved the convenience of skull-based sex estimation, making the process more efficient and straightforward for users. A range of sex estimation software has become available. This includes non-metric tools like CroWalker [114] and MorphoPASSE [131], alongside traditional metric software such as SexEst [43], Fordisc [115], and KKU Sex Estimation [132], as well as GM applications like SkullProfiler [46] and 3D-ID [133]. This variety of software ensures that sex estimation from skulls is more comprehensive, offering practitioners a choice of tools suited to their specific needs. Furthermore, given the uniqueness of different populations, future software updates are expected to incorporate data from diverse worldwide populations to enhance the applicability and accuracy of these tools.

## Limitations of the scoping review

Due to the multiple variables involved in sex estimation studies, such as population, sample size, modality, cranial traits, and prediction models, it becomes extremely challenging to precisely control experimental variables to assess the impact of one single factor. Therefore, our review only offers a rough outline of the efficacy differences among different technologies in sex estimation. Moreover, while DNA testing can achieve 100% accuracy in sex classification, our study doesn't delve into genetic techniques. Conversely, our focus is primarily on the metric and non-metric methods widely employed in archaeology and anthropology. These methods not only assist DNA technique in initial sex estimation, but they also possess the unique ability to analyze cranial feature differences among various populations, thus offering valuable insights into human biological diversity.

## Conclusions

Sex estimation has an extensive practical value in forensic anthropology. Our study employed modalities, methods, and models as a framework to investigate the strengths and limits of various technologies in skull-based sex estimation. Our purpose is to facilitate practitioners' rapid comprehension of the components of sex estimation and empower them to select cranial features and research methodologies that optimally address their practical needs in the identification process. Additionally, in instances where extensive scientific research may be constrained, we summarized several software tools that have the potential to aid forensic investigators in efficiently and accurately performing sex estimation on skulls.

Through a synthesis of recent literature, we have discovered that with the development of medical 3D imaging, the efficiency of skull-based sex classification has been improved. However, despite these advancements, several limitations persist in this field, necessitating future attention. For instance, improving the generalization ability of models to handle skull data from diverse populations, reducing the technological costs to enable wider application scenarios, and validating established sex prediction models on independent datasets to ensure their robustness and applicability.

Looking ahead, research in sex estimation is expected to achieve breakthroughs in several areas. With the application of big data and AI techniques, it is anticipated that the performance of multiple models will be evaluated using different performance indicators, thereby enhancing the reliability of sex estimation outcomes. Furthermore, the combination of 3D imaging and computer vision techniques might capture more detailed information about skulls, providing finer data support for sex estimation.

## Supporting information

**S1 Table. PRISMA-ScR checklist.**
(DOCX)

**S2 Table. Characteristics of the included articles.**
(DOCX)

## Author Contributions

**Conceptualization:** Xindi Wang, Yuan Li.

**Data curation:** Xindi Wang, Guihong Liu, Yazi Zheng.

**Formal analysis:** Qiushuo Wu.

**Funding acquisition:** Yuan Li.

**Methodology:** Guihong Liu.

**Supervision:** Feng Song, Yuan Li.

**Visualization:** Qiushuo Wu.

**Writing – original draft:** Xindi Wang.

**Writing – review & editing:** Feng Song.

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
