## [Decision Letter · Decision Letter 0]

3 Sep 2024

PONE-D-24-32582Sex estimation techniques based on skulls in forensic anthropology: a scoping reviewPLOS ONE

Dear Dr. Li,

Thank you for submitting your manuscript to PLOS ONE. After careful consideration, we feel that it has merit but does not fully meet PLOS ONE’s publication criteria as it currently stands. Therefore, we invite you to submit a revised version of the manuscript that addresses the points raised during the review process.

We look forward to receiving your revised manuscript.

Kind regards,

Rijen Shrestha, M.D.

Academic Editor

PLOS ONE

Journal Requirements: When submitting your revision, we need you to address these additional requirements. 1. Please ensure that your manuscript meets PLOS ONE's style requirements, including those for file naming. The PLOS ONE style templates can be found at https://journals.plos.org/plosone/s/file?id=wjVg/PLOSOne_formatting_sample_main_body.pdf and https://journals.plos.org/plosone/s/file?id=ba62/PLOSOne_formatting_sample_title_authors_affiliations.pdf 2. Thank you for stating the following financial disclosure: "This study was supported by the National Natural Science Foundation of China (No. 82202079) and Natural Science Foundation of Sichuan Province (No. 2022NSFSC1403), both awarded to Ms Yuan Li." Please state what role the funders took in the study.  If the funders had no role, please state: ""The funders had no role in study design, data collection and analysis, decision to publish, or preparation of the manuscript."" If this statement is not correct you must amend it as needed. Please include this amended Role of Funder statement in your cover letter; we will change the online submission form on your behalf. 3. PLOS requires an ORCID iD for the corresponding author in Editorial Manager on papers submitted after December 6th, 2016. Please ensure that you have an ORCID iD and that it is validated in Editorial Manager. To do this, go to ‘Update my Information’ (in the upper left-hand corner of the main menu), and click on the Fetch/Validate link next to the ORCID field. This will take you to the ORCID site and allow you to create a new iD or authenticate a pre-existing iD in Editorial Manager. 4. Please review your reference list to ensure that it is complete and correct. If you have cited papers that have been retracted, please include the rationale for doing so in the manuscript text, or remove these references and replace them with relevant current references. Any changes to the reference list should be mentioned in the rebuttal letter that accompanies your revised manuscript. If you need to cite a retracted article, indicate the article’s retracted status in the References list and also include a citation and full reference for the retraction notice. 

Reviewers' comments:

Reviewer's Responses to Questions

**Comments to the Author**

1. Is the manuscript technically sound, and do the data support the conclusions?

Reviewer #1: Yes

Reviewer #2: Yes

2. Has the statistical analysis been performed appropriately and rigorously? 

Reviewer #1: N/A

Reviewer #2: Yes

3. Have the authors made all data underlying the findings in their manuscript fully available?

Reviewer #1: Yes

Reviewer #2: Yes

4. Is the manuscript presented in an intelligible fashion and written in standard English?

Reviewer #1: No

Reviewer #2: Yes

5. Review Comments to the Author

Reviewer #1: Dear Authors,

I would like to congratulate you on conducting a scoping review on the modern techniques being used in sex estimation using skull. I have a few suggestions/comments on the article that are as follows:

Abstract:

Line 18: The statement, "An emerging body of techniques have been examined by recent studies for sex estimation using human skulls" seems incoherant and should be rephrased.

Introduction:

Line 31: The first line of the introduction seems incoherant and should be rephrased

Line 36-36: Replace the phrase 'a diverse factor' to 'diverse factors'

Lines 41-42: Only a single paper has been cited for each of the anatomical structures used for sex estimation. The authors should cite multiple papers.

Lines 42-43: Please provide citation for referral of pelvis as the best indicator of sex

Line 56: The quality of Figure 1 is poor and the authors should submit an image with a DPI of greater than 800

Line 62: Replace the word 'learning' with 'analysis'

Materials and Methods:

Line 170-171: The sentence reads incoherantly and has misspelled word (Chinese).

Line 193: Rephrase to 'After screening the articles based on their titles and abstracts, 507 articles were excluded'

Line 198: The figure is blurred and the authors are recommended to use an image with at least 800 DPI

Line 203-204: The authors have mentioned Iran to be a part of Europe. This is factually incorrect as the Republic of Iran is a part of West Asia

Line 205-206: ... and in American countries ....

Line 206-207: State in which countries were the ref. no. 22 and 93 conducted. Include that as a part of the european section mentioned in lines 203-204

Synthesis of Results:

Line 209: The figure is blurred and needs to be resubmitted in a higher DPI (>800)

Line 219: Same as the comments for all other figures.

Line 238: Same as the comments for all other figures.

Line 264: Same as the comments for all other figures.

Line 280: Please provide the long forms of abbreviations mentioned

Overall, the article requires corrections in grammar and language, preferably done by someone who is a native english speaker. Other than that, I commend the authors on completing an informative scoping review on sex estimation using the skull, and suggest them to incorporate the aforementioned changes.

Reviewer #2: Thank you for the opportunity to review this interesting article. I have a few questions, which are listed below:

The present study is a very elaborative, systematic review of sex estimation techniques based on skulls in Forensic Anthropology. Now my query is what contribution it has made in the identification process. Please add one paragraph related to it.

Page no. 10, 1st Paragraph, line no. 195: Remove the in from “resulting in in” is written twice.

Page no. 10, Modalities: please specify why CT is the most preferred imaging modality.

6. PLOS authors have the option to publish the peer review history of their article (what does this mean?). If published, this will include your full peer review and any attached files.

Reviewer #1: **Yes: **Rutwik Shedge

Reviewer #2: No

---

## [Author Response · Author response to Decision Letter 0]

12 Sep 2024

Manuscript ID: PONE-D-24-32582

Article Type: Research Paper

Title: Sex estimation techniques based on skulls in forensic anthropology: a scoping review 

Dear Dr Rijen Shrestha and Reviewers,

Thank you very much for giving us a chance to revise our manuscript. The reviewer’s comments are valuable and very helpful for improving our research paper. We have carefully read all comments and have tried our best to revise the manuscript as per reviewers’ suggestions, which we hope to meet with acceptance requirements. Reviewer # 1 comments have been highlighted in yellow, and Reviewer # 2 comments have been highlighted in blue in the manuscript. 

Response to Journal Requirements

l Please ensure that your manuscript meets PLOS ONE's style requirements, including those for file naming.

Response: We have reviewed the manuscript to ensure compliance with PLOS ONE's style requirements, including file naming conventions. 

l Thank you for stating the following financial disclosure. Please state what role the funders took in the study. 

Response: Thank you for the prompt. The funder had no role in study design, data collection and analysis, decision to publish, or preparation of the manuscript. We have included this amended Role of Funder statement in our cover letter.

l Please ensure that the corresponding author has an ORCID iD and that it is validated in Editorial Manager. 

Response: The corresponding authors’ ORCID iDs have been created and validated in Editorial Manager as per PLOS requirements.

l Please review your reference list to ensure that it is complete and correct.

Response: The revised reference list is now complete and correct, including newly added citations as presented in our response to reviewers. The citation format has been adjusted to comply with PLOS ONE's requirements, and the corresponding reference numbers in Supplementary Table 2 have also been updated accordingly.

Response to Reviewer #1

Reviewer #1: 

l I would like to congratulate you on conducting a scoping review on the modern techniques being used in sex estimation using skulls. I have a few suggestions/comments on the article that are as follows.

Response: Thank you very much for your support of our work. You have provided us with valuable advice on how to improve the quality of this paper. We have used your comments to revise the manuscript and have attached a point-by-point response to your comments.

Q1: Abstract: Line 18: The statement, "An emerging body of techniques have been examined by recent studies for sex estimation using human skulls" seems incoherent and should be rephrased.

Response: Thank you for pointing out the incoherence in Line 18 of the Abstract. You are correct that the original phrasing could be improved for clarity. In response to your suggestion, we have revised the sentence as follows. 

Original sentence: An emerging body of techniques have been examined by recent studies for sex estimation using human skulls.

Modified sentence (Page 2, Lines 15-16): Recent studies have investigated a growing range of techniques for estimating sex from human skulls.

Q2: Introduction: Line 31: The first line of the introduction seems incoherent and should be rephrased.

Response: Thank you for bringing this to our attention. The corresponding changes are shown below.

Original sentence: Forensic anthropology techniques are expected to develop the biological profile of human remains, which may provide investigation directions and narrow down the search to a specific group.

Modified sentence (Page 2, Lines 30-32): Forensic anthropology often involves estimating sex, ancestry, age, and height from the skeletal remains of unknown individuals. This estimated summary is referred to as the biological profile, which is subsequently used to compare with missing person records to achieve personal identification [1].

1. Christensen AM, Passalacqua NV, Bartelink EJ. Chapter 1 - Introduction. In: Christensen AM, Passalacqua NV, Bartelink EJ, editors. Forensic Anthropology. San Diego: Academic Press; 2014. p. 1-17.

Q3: Introduction: Line 36-36: Replace the phrase 'a diverse factor' to 'diverse factors'.

Response: We feel sorry for our carelessness. In our resubmitted manuscript, we have corrected 'a diverse factor' into 'diverse factors'. Thank you for your reminder.

Q4: Introduction: Lines 41-42: Only a single paper has been cited for each of the anatomical structures used for sex estimation. The authors should cite multiple papers.

Response: Dear reviewer, thank you for your valuable comments. We have included additional citations for each anatomical structure in the mentioned lines. These articles are as follows:

8. Ives G, Johns SE, Deter C. Sexual dimorphism of pelvic scarring: A new method of adult biological sex estimation. J Forensic Sci. 2024;00:1-13. doi: 10.1111/1556-4029.15587.

9. Zhang K, Zhan M, Deng L, Qiu LR Deng ZH. Estimation of stature and sex from pelvic measurements in a Chinese population. Aust J Forensic Sci. 2020;52(4):406-16. doi: 10.1080/00450618.2018.1541193.

11. Partido NM, Fombuena ZI, Borja MEA, Alemán AI. Discriminant functions for sex estimation using the rib necks in a Spanish population. Int J Legal Med. 2021;135(3):1055-65. doi: 10.1007/s00414-021-02537-8.

12. Kubicka AM, Piontek J. Sex estimation from measurements of the first rib in a contemporary Polish population. Int J Legal Med. 2016;130(1):265-72. doi: 10.1007/s00414-015-1247-6.

14. Koşar Mİ, Uğuz Gençer C, Tetiker H, Yeniçeri İÖ, Çullu N. Sex and stature estimation based on multidetector computed tomography imaging measurements of the sternum in Turkish population. Foren Imag. 2022;28:200495. doi: 10.1016/j.fri.2022.200495.

15. Sehrawat JS. Sex estimation from discriminant function analysis of clavicular and sternal measurements: a forensic anthropological study based on examination of two bones of Northwest Indian subjects. Aust J Forensic Sci. 2018;50(1):20-41. doi: 10.1080/00450618.2016.1188986.

17. Karaca AM, Senol E, Eraslan C. Evaluation of the usage of the cervical 7th vertebra in sex estimation with measurements on computerized tomography images. Legal Med-Tokyo. 2023;62:102220. doi: 10.1016/j.legalmed.2023.102220.

18. Azofra-Monge A, Alemán Aguilera I. Morphometric research and sex estimation of lumbar vertebrae in a contemporary Spanish population. Forensic Sci Med Pat. 2020;16(2):216-25. doi: 10.1007/s12024-020-00231-6.

20. Bozdag M, Er A, Kranioti E, Basa CD, Oztop B, Kacmaz E, et al. Sex estimation in a modern Turkish population using the clavicle: a computed tomography study. Aust J Forensic Sci. 2022;54(2):187-98. doi: 10.1080/00450618.2020.1781255.

21. Hisham S, Lai PS, Ibrahim MA, Zainun KA. Sex estimation using post-mortem computed tomographic images of the clavicle in a Malaysian population. Legal Med-Tokyo. 2024;71:102500. doi: 10.1016/j.legalmed.2024.102500.

23. Bertsatos A, Garoufi N, Chovalopoulou M-E. Advancements in sex estimation using the diaphyseal cross-sectional geometric properties of the lower and upper limbs. Int J Legal Med. 2021;135(3):1035-46. doi: 10.1007/s00414-020-02437-3.

24. Phuwadon D. Sex estimation from upper limb bones in a Thai population. Anat Cell Biol. 2020;53(1):36-43. doi: 10.5115/acb.19.179.

Q5: Introduction: Lines 42-43: Please provide citation for referral of pelvis as the

best indicator of sex.

Response: As suggested by the reviewer, we have added more references to support the idea of the pelvis as the best indicator of sex. These articles are as follows:

7. Cao YJ, Ma YG, Yang XT, Xiong J, Wang YH, Zhang JH, et al. Use of deep learning in forensic sex estimation of virtual pelvic models from the Han population. Foren Sci Res. 2022;7(3):540-9. doi: 10.1080/20961790.2021.2024369.

25. d’Oliveira Coelho J, Curate F. CADOES: An interactive machine-learning approach for sex estimation with the pelvis. Forensic Sci Int. 2019;302:109873. doi: 10.1016/j.forsciint.2019.109873.

26. Stan E, Muresan C-O, Dumache R, Ciocan V, Ungureanu S, Costachescu D, et al. Sex Estimation from Computed Tomography of Os Coxae—Validation of the Diagnose Sexuelle Probabiliste (DSP) software in the Romanian population. Appl Sci-Basel. 2024;14(10):4136. doi:10.3390/app14104136.

Q6: Introduction: Line 56: The quality of Figure 1 is poor and the authors should submit an image with a DPI of greater than 800.

Response: Thank you for the feedback. We have optimized Figure 1's DPI to the maximum extent possible within PLOS ONE's specified DPI range (300-600 DPI) to ensure clarity. The updated figure has been resubmitted to the PLOS ONE submission system.

Q7: Introduction: Line 62: Replace the word 'learning' with 'analysis'.

Response: Thank you for your reminder. We have replaced the word 'learning' with 'analysis' in Line 67 (The change in line numbers is due to revisions made to the manuscript.).

Q8: Materials and Methods: Line 170-171: The sentence reads incoherently and has misspelled word (Chinese).

Response: We sincerely thank you for your careful reading. We have corrected the misspelled word and changed the sentence as follows.

Original sentence: Keywords related to the study aim and included in the search string were: (“sex" OR "gender") AND ("determination" OR "estimation" OR "prediction") AND "skull". The English and Chinses language and time interval of publication, from January 2020 to February 2024, were applied as filters.

Modified sentence (Page 9, Lines 177-180): The search string included keywords related to the study's aim: ('sex' OR 'gender') AND ('determination' OR 'estimation' OR 'prediction') AND 'skull'. Filters were applied to limit the search results to articles published between January 2020 and February 2024 in English and Chinese.

Q9: Materials and Methods: Line 193: Rephrase to 'After screening the articles based on their titles and abstracts, 507 articles were excluded'.

Response: Thank you for the suggestion. We have rephrased the sentence on Line 193 to improve clarity. 

Original sentence: After titles and abstracts were screened, 507 articles were removed.

Modified sentence (Page 10, Lines 202-203): After reviewing the titles and abstracts, we excluded 507 articles from further consideration.

Q10: Materials and Methods: Line 198: The figure is blurred and the authors are recommended to use an image with at least 800 DPI.

Response: Thank you for your comments. We have replaced Figure 2 with a higher-resolution image to ensure clarity. The updated figure has been resubmitted to the PLOS ONE submission system.

Q11: Materials and Methods: Line 203-204: The authors have mentioned Iran to be a part of Europe. This is factually incorrect as the Republic of Iran is a part of West Asia.

Response: We were really sorry for our careless mistakes. You are absolutely correct in pointing out that the Republic of Iran is located in West Asia. We have corrected the geographical location of Iran and revised the relevant reference numbers in our manuscript.

Original sentence: The majority of studies on sex estimation have been conducted on Asian populations, with India [43-56] leading the way, followed by Turkey [20, 57-63] and China [64-66]. 

Modified sentence (Page 11, Lines 212-213): The majority of studies on sex estimation were conducted on Asian populations, with India [58-71] leading the way, followed by Turkey [35, 72-78], Iran [79-84], and China [85-87].

Q12: Materials and Methods: Line 205-206: ... and in American countries ...

Response: Thank you for your careful checks. We are sorry for our carelessness. We have revised the sentence to make it grammatically accurate.

Original sentence: Several studies were conducted on African countries including Nigeria [82-84] and Egypt [85, 86], American countries including Brazil [87, 88] and the United States [89].

Modified sentence (Page 11, Lines 217-219): Several studies were conducted in African countries, including Nigeria [98-100] and Egypt [101, 102], and in American countries, including Brazil [103, 104] and the United States [105].

Q13: Line 206-207: State in which countries were the ref. no. 22 and 90 conducted. Include that as a part of the European section mentioned in lines 203-204.

Response: We appreciate your pointing this out to us. According to your suggestion, we have clarified the countries of the study in ref. No. 90 (now revised to 97) within the European section. Unfortunately, ref. No. 22 (revised to 37) failed to specify the countries from which its samples originate.

Original sentence: Additionally, one study concerned Caucasians [22] and one concerned 4 distinct populations in Western Europe [90].

Modified sentence (Page 11, Lines 213-216): In Europe, notable contributions came from Bosnia [88-92] and Bulgaria [36, 93-96], while a single study encompassed four distinct Western European populations: Belgium, Switzerland, France, and Portugal [97].

Modified sentence (Page 11, Lines 219-220): Notably, one study concerned Caucasians, but the countries where this study was conducted were not specified [37].

Q14: Synthesis of Results: Line 209: The figure is blurred and needs to be resubmitted in a higher DPI (>800).

Line 219: Same as the comments for all other figures.

Line 238: Same as the comments for all other figures.

Line 264: Same as the comments for all other figures.

Response: Thank you for highlighting the issue with the blurred figures. We apologize for any inconvenience caused by this oversight. We have optimized all figures to a higher DPI within PLOS ONE's specified DPI range (300-600 DPI) to improve clarity. The updated figures have been resubmitted to the PLOS ONE submission system.

Q15: Line 280: Please provide the long forms of abbreviations mentioned.

Response: Thank you for your reminder. We are sorry for our carelessness. In the revised manuscript, we have added the long form 'random forest' to the acronym RF. DFA and SVM have been defined where they first appear in the manuscript (specifically, on Page 7, Lines 139 and 146). Furthermore, we have provided a list of abbreviations in Supplementary Table 2 to facilitate the reading of our manuscript.

Original sentence: Additionally, SexEst integrates various classification models, including DFA, RF, SVM, and gradient boosting (GB), allowing users to choose the model that best suits their data.

Modified sentence (Page 15, Line 301): Additionally, SexEst integrates various classification models, including DFA, SVM, random forest (RF), and gradient boosting (GB), allowing users to choose the model that best suits their data.

Q16: Overall, the article requires corrections in grammar and language, preferably done by someone who is a native English speaker.

Response: Thank you for your suggestion. We have worked on both language and readability and have also invited a native English speaker to help revise the grammar errors. We have not listed the changes here but have highlighted them in red. We sincerely hope that these revisions will meet with your approval and satisfaction.

l Other than that, I commend the authors on completing an informative scoping review on sex estimation using the skull, and suggest them to incorporate the aforementioned changes.

Response: Thank you for your kind comments and valuable suggestions. We have incorporated the changes you suggested.

Response to Reviewer #2

Reviewer #2: 

l Thank you for the opportunity to review this interesting article. I have a few questions, which are listed below.

Response: We greatly appreciate your commendation on our review, as well as your constructive suggestions for improvement. We have carefully considered and incorporated your suggestions into our manuscript. All the changes are highlighted in blue in the revised manuscript for your comments. Specific to each point, the revision notes are given as follows.

Q1: The present study is a very elaborative, systematic review of sex estimation techniques based on skulls in Forensic Anthropology. Now my query is what contribution it has made in the identification process. Please add one paragraph related to it.

Response: Dear reviewer, we appreciate the opportunity to clarify the contributions of our study for forensic identification. We have conducted a comprehensive literature review to present and discuss various techniques for estimatin

---

## [Editor Report · Decision Letter 1]

24 Sep 2024

Sex estimation techniques based on skulls in forensic anthropology: a scoping review

PONE-D-24-32582R1

Dear Dr. Li,

We’re pleased to inform you that your manuscript has been judged scientifically suitable for publication and will be formally accepted for publication once it meets all outstanding technical requirements.

Kind regards,

Rijen Shrestha, M.D.

Academic Editor

PLOS ONE
---

## [Editor Report · Acceptance letter]

29 Sep 2024

PONE-D-24-32582R1 

PLOS ONE

Dear Dr. Li, 

I'm pleased to inform you that your manuscript has been deemed suitable for publication in PLOS ONE. Congratulations! Your manuscript is now being handed over to our production team.

Kind regards, 

on behalf of

Dr. Rijen Shrestha 

Academic Editor

PLOS ONE